# Optimistic Asynchrony Control: Achieving Synchronous Convergence With Asynchronous Throughput for Embedding Model Training

**Roger Waleffe** [1]  **Jason Mohoney** [1]

## Abstract

Modern embedding-based machine learning (ML) models can contain hundreds of gigabytes of parameters, often exceeding the capacity of GPU hardware accelerators critical for training. One solution is to use a mixed CPU-GPU setup, where embedding parameters are stored in CPU memory and subsets are repeatedly transferred to the GPU for computation. In this setup two training paradigms exist: synchronous training and asynchronous training. In the former, batches are transferred one by one, leading to low throughput but fast model convergence. In contrast, during asynchronous training batches are transferred in parallel, allowing for more batches to be processed per unit time. Asynchronous training, however, can effect model quality due to concurrent batches which access the same model parameters leading to stale updates. In this work, we present Optimistic Asynchrony Control, a method for allowing asynchronous batch processing while ensuring model equivalence to a synchronous training execution. Our method is inspired by Optimistic Concurrency Control used in database systems. The main idea is to allow parallel processing and transfer of batches from the CPU to the GPU, but to validate each batch on the GPU before the model is updated to ensure that it has the correct values—the values it would have had if batches were processed and transferred one by one. We show that OAC achieves the best of both worlds, retaining the convergence of synchronous training while matching the throughput of asynchronous ML. This allows OAC to achieve the best time-to-accuracy of the three methods for mixed CPU-GPU embedding model training.

[1]University of Wisconsin-Madison. Correspondence to: Roger Waleffe <waleffe@wisc.edu>.

Accepted to the Workshop on Advancing Neural Network Training at International Conference on Machine Learning (WANT@ICML 2024).

## 1. Introduction

Machine learning (ML) models have shown great success across many disciplines, including computer vision (He et al., 2016; Huang et al., 2017; Dosovitskiy et al., 2020), natural language processing (Devlin et al., 2018; Brown et al., 2020; Radford et al., 2019; OpenAI, 2023), and graph learning (Mohoney et al., 2021; Chami et al., 2021; Kipf & Welling, 2016; Shang & Chen, 2021; Derrow-Pinion et al., 2021; Jumper et al., 2021). As these models have become more prevalent, two trends have emerged with respect to ML training: First, practitioners wish to train models with an increasingly large number of parameters, whether for increased accuracy (Sevilla et al., 2022), or because they wish to train on more data (e.g. larger graphs) (Mohoney et al., 2021). Second, hardware accelerators (GPUs) have become (nearly) mandatory for computation. Interestingly, these two trends conflict with each other. Modern ML models can contain many hundreds of gigabytes of data, far exceeding the memory capacity of accelerators (max memory 10s of gigabytes). Thus, it has become common to use multiple layers of the memory hierarchy for ML training. One such approach, commonly used for embedding-based ML models (Mohoney et al., 2021; Zheng et al., 2020; Lerer et al., 2019), and the focus of this work, is mixed CPU-GPU training (Dong et al., 2021).

In mixed CPU-GPU training, (some) ML model parameters are stored in CPU memory. Batches (subsets) of these parameters are then read and transferred to a GPU for computation where they are updated as part of the learning process. The updated values are then transferred back to the CPU and written to the CPU memory. Continuous repetition of this procedure eventually converts the initial parameters into the final learned ML model. Conventional ML algorithms proceed according to a *synchronous* manner: One batch is transferred and updated at a time. A subsequent batch does not begin until the previous batch has completed. Synchronous ML is the default in many training frameworks but it suffers from low throughput because the accelerator is idle during batch preparation and transfer (Mohoney et al., 2021). Thus, systems are increasingly supporting *asynchronous* training. In this case, batches are allowed to run in parallel to increase throughput. Multiple batches are

prepared and transferred at once so that the GPU continually has batches to process.

While asynchronous ML training can improve throughput in the mixed CPU-GPU setting, it can suffer from suboptimal convergence—requiring a greater number of batches (updates) to achieve a given model accuracy than the synchronous alternative, or failing to reach the given accuracy altogether. The issue with asynchronous training is that multiple concurrent batches may read and update the same parameters (we say these batches *overlap*). Only the last write to the CPU memory will persist, leading to lost updates for some batches. The overall effect this has on model training can vary substantially. It's often best to compare synchronous and asynchronous training in terms of *time-to-accuracy*, i.e. how long (wall clock training time) does it take for a model to reach a given quality. For example, if asynchronous training increases throughput by a factor of two, it can process twice as many batches as synchronous training per unit time. However, if it also requires twice as many batches to converge to the desired accuracy, then the overall training time of the two methods will be the same. In general, the optimal method for time-to-accuracy depends on the application. Asynchronous training can perform well when the overlap between concurrent batches is low (Niu et al., 2011), but can also result in significantly degraded convergence, potentially rendering it unusable.

The question we ask in this work is whether we can *ensure synchronous convergence* while maintaining *asynchronous throughput*. Such a method would result in the best time-to-accuracy for mixed CPU-GPU ML training. The main difficulty is how to handle concurrent batches which overlap.

The solution we propose is motivated by database concurrency control which solves an analogous challenge. To increase throughput, databases allow multiple transactions to run in parallel, but they ensure that the final result is equivalent to a serial execution of each transaction. Initial (pessimistic) approaches relied on locking of data objects to prevent two simultaneous threads from modifying the same records. Newer approaches, however, utilize a different technique. Optimistic methods for concurrency control, termed OCC, hope that conflicts between two concurrent transactions will not occur (Kung & Robinson, 1981; Tu et al., 2013). Each transaction is allowed to proceed at the same time, but each must track its reads and write its modifications to thread local storage. Before committing its local writes to the database, a transaction must validate that no concurrent transaction's writes overlapped with its read set. If validation fails, two transactions accessed the same data concurrently and one transaction must abort to prevent an inconsistent database state.

Here we propose **Optimistic Asynchrony Control (OAC)** to ensure that asynchronous, parallel processing of batches

in mixed CPU-GPU ML training is equivalent to a synchronous, one-by-one execution. Rather than adopting a pessimistic approach and preventing batches which overlap from running in parallel, we take an optimistic approach and allow all batches to run concurrently but *validate* parallel batches against each other for overlap. Unlike in conventional OCC where overlapping transactions require aborts, in OAC we show how batches which access the same parameters can be updated at validation time to have the correct (according to a synchronous order) values.

The key contributions of OAC are as follows. First, we highlight that the order in which batches pass through the GPU computation step defines a one by one order over batches. In OAC, we ensure that asynchronous training results in the same updates to model parameters as synchronous training would have produced according to this order. Second, we introduce timestamps to each parameter to track its most recent update and allow us to easily decide which value to accept for a parameter in the presence of multiple options. Finally, we add an on GPU parameter cache which tracks the parameter sets of concurrent batches. This allows us to validate that a batch has the correct parameter values just before it enters computation to produce model updates.

We implement OAC in the graph learning system Marius (Mohoney et al., 2021; Waleffe et al., 2023) and evaluate our method on the link prediction ML task. Link prediction is a natural first application of OAC, as it requires learning a vector (embedding) with hundreds of parameters for every node in a graph. Modern graphs have hundreds of millions of nodes resulting in total model sizes exceeding GPU memory capacities. We show that OAC results in identical convergence to synchronous training and identical throughput to asynchronous training. This allows OAC to achieve the best of both worlds, resulting in the fastest time-to-accuracy. OAC reaches the same accuracy as synchronous training up to $3\times$ faster, and consistently achieves higher accuracy that asynchronous training without sacrificing throughput.

The rest of this paper is organized as follows. In Section 2 we discuss synchronous and asynchronous ML training in more detail, providing an example to highlight the difference. We also discus Marius and the graph learning task we use for evaluation. In Section 3 we describe OAC for an abstract mixed CPU-GPU ML training task in detail. In Section 4 we present our evaluation of OAC versus standard asynchronous and synchronous training on the graph link prediction task. Final we conclude and present future directions in Section 5.

## 2. Preliminaries

In this section, we begin by discussing in more detail background information which is useful for understanding later

parts of the paper. In particular, we highlight the difference between synchronous and asynchronous training by introducing a running example.

## 2.1. Synchronous vs. Asynchronous Training

As discussed above, in mixed CPU-GPU training model parameters are stored in CPU memory (Figure 1). There are two main learning paradigms for updating these parameters to their final state. In **synchronous training** (Figure 1a) a subset of these parameters, called a *batch*, is first read from CPU memory and then transferred to the GPU. Once on the GPU, the ML model can perform its training task on these parameters to compute updated values. These updated values are then transferred back to the CPU and subsequently written to CPU memory. Each step in this process happens sequentially, waiting for the previous step to finish. Repeating this process eventually leads to the final learned ML model. Note that each parameter may need to be updated many times during training. In the example in Figure 1a, a batch reads three parameters $\{a, c, f\}$ and updates them to $\{a', c', f'\}$. A second batch will wait to start until the first batch is completed. This second batch may read, for example, parameters $\{a', b, h\}$ and update them to $\{a'', b', h'\}$. Notice that the second batch is guaranteed to see the update from $a$ to $a'$ from the first batch. This is the strength of synchronous training—all updates from one batch are seen by all subsequent batches—and leads to fast convergence (the model requires fewer batches to learn). However, the drawback of this paradigm is low throughput (fewer batches are processed per unit time), as the GPU is idle while it waits for a batch to be read, transferred, and written (steps 1, 2, 4, and 5 in Figure 1a).

The second learning paradigm is **asynchronous training** which seeks to address the low throughput problem of synchronous training. Instead of reading and transferring one batch at a time, here multiple batches are prepared and sent to the GPU in parallel. The goal is to keep the GPU busy: as soon as it finishes computation on one batch, another batch is waiting in GPU memory for processing. This increases throughput, allowing more batches to be processed per unit time. Asynchronous training is generally implemented using multiple threads and queues (Mohoney et al., 2021). CPU reader threads continually prepare and transfer batches concurrently and push them to an on GPU input queue. The GPU can then repeatedly read off batches from this data structure. After updating a batch, the GPU puts the new values in an on GPU output queue. Multiple CPU writer threads continually grab batches from this output queue to transfer and write the results back to CPU memory.

While asynchronous training increase throughput, it can negatively affect convergence. This is because the synchronous property described above no longer holds—it is no longer the case that all updates from one batch are seen by all subsequent batches. Consider the example in Figure 1b. Two batches are prepared in parallel. One reads $\{a, c, f\}$ and another reads $\{a, b, h\}$. Notice that these batches *overlap*, i.e. they both read the same parameter, $a$ in this case. The updates from the first batch ($\{a, c, f\} \rightarrow \{a', c', f'\}$) have not made it back to CPU memory before the second batch ($\{a, b, h\}$) is started. This is in contrast with the synchronous setting. When both batches reach the GPU, one will be pulled of the input queue and processed first, followed by the second batch. In this example, both batches will update $a$ to their own version of $a'$ (we differentiate each version of $a'$ by a $a'(1)$ or a $a'(2)$ when needed). When these batches subsequently write their updates back to CPU memory, only the last write will persist, for example $a'(1)$. The processing of these two batches in the asynchronous setting has resulted in a different model parameter state $\{a'(1), b', c', d, e, f', g, h'\}$ than the result of processing these batches one at a time in the synchronous setting $\{a'', b', c', d, e, f', g, h'\}$. The consequence is that asynchronous training may not be able to achieve the same final model accuracy as synchronous training, or it may converge more slowly, requiring more batches to be processed to reach a given model quality.

Note that the problem with asynchronous training occurs due to batches which concurrently access the same parameters. The frequency and number of these conflicts depends on many factors including: the batch size, the total number of parameters, the parameter access pattern across batches, the number of reader and writer threads, the queue sizes, etc. It is possible to tune some of these parameters to reduce conflicts and improve the convergence of asynchronous training, but changing these parameters could also affect model accuracy and throughput. For example, reducing the number of reader threads will reduce the number of concurrent batches and reduce conflicts, but may also reduce throughput. Other factors are hard to change. The parameter access pattern can generally be assumed to be random as it is nearly always preferred to shuffle training examples during ML training for improved accuracy (a group of training examples define the parameters required for a given batch).

To summarize synchronous versus asynchronous training we draw an **analogy** to database systems. A batch in our setting is roughly equivalent to a transaction in a database. Both read and write values to a shared state. Synchronous training can be viewed as a serial schedule of transactions. Each transaction (batch) performs its reads and writes one at a time. Asynchronous training on the other hand is analogous to an interleaving schedule of transactions where reads and writes from different batches happen in parallel. *They key difference, however, is that database systems include concurrency control mechanisms to ensure that parallel execution is equivalent to a serial execution. No such analogous*

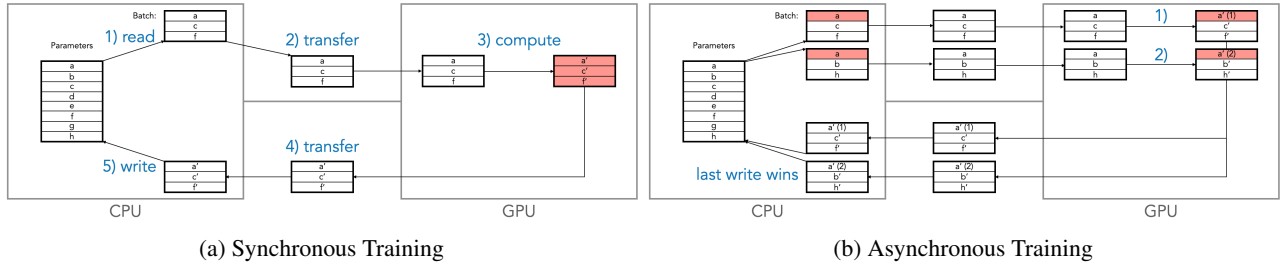

(a) Synchronous Training                                    (b) Asynchronous Training

*Figure 1.* Graphical comparison of synchronous versus asynchronous mixed CPU-GPU training.

*mechanism exists for asynchronous training.*

## 2.2. Marius and Graph Learning

In Section 3 we present our method for ensuring asynchronous training is equivalent to synchronous training for a general mixed CPU-GPU ML training setup. In Section 4 we evaluate an initial implementation of this algorithm in the graph learning system Marius. Here we provide additional information about this system and ML learning problem. Marius is a system for learning large scale graph embeddings on a single machine. It supports both synchronous and asynchronous mixed CPU-GPU training. The primary learning task is *link prediction*. Given a graph with a set of nodes $V$ and a set of edges $E$, the task is to learn an embedding (vector) of parameters for each node $n \in V$, such that these node embeddings can recover the edges of the graph (through mathematical operations). Trained embeddings can then be used to predict the existence of new edges or to filter erroneous edges already present in the graph. More details can be found here (Mohoney et al., 2021).

## 3. Optimistic Asynchrony Control

In this section, we present Optimistic Asynchrony Control (OAC), a method for allowing asynchronous parallel processing and transfer of batches in mixed CPU-GPU training while guaranteeing that the final model state is equivalent to one which would have been generated by synchronous training. The key idea is to *validate* each batch against other concurrent batches to detect parameter overlap and then to ensure that each batch uses the correct parameter values for computation. Here, the correct parameters refer to the values a batch would have read if it had been processed in some synchronous order.

## 3.1. The Synchronous Order

In OAC, we would like to ensure asynchronous training produces updates to the model state equivalent to some synchronous training execution. The first question is then: what synchronous order should we validate against? Recall from Section 2.1 that in asynchronous training, batches are read

and transferred in parallel from the CPU to an on GPU input queue. The GPU then reads batches one by one from this queue for computation (the GPU computation can be viewed as a critical section). We will call this sequential order over batches the *computation order*. For example, in Figure 1b, first $\{a, c, f\}$ is updated to $\{a'(1), c', f'\}$, and then $\{a, b, h\}$ is updated to $\{a'(2), b', h'\}$. This is the critical observation for OAC. We will ensure that asynchronous training produces the same updates as synchronous training would have produced had it processed batches in this computation order. Note that even though batches are processed by the GPU sequentially in asynchronous training, it is not equivalent to synchronous training by default. Since batches $\{a, c, f\}$ and $\{a, b, h\}$ were read and transferred to the GPU concurrently, they share the same value for parameter $a$. Sequential training with the given computation order would have processed batch $\{a, c, f\} \rightarrow \{a', c', f'\}$, then waited for these updates to reach the CPU, and only then started processing the second batch. Thus, the second batch would have seen the update from $a \rightarrow a'$ and computed $\{a', b, h\} \rightarrow \{a'', b', h'\}$ as described in Section 2.1.

## 3.2. Equivalence to The Synchronous Order

Given the computation order defined in Section 3.1, we now seek to ensure that asynchronous training is equivalent to synchronous training under this sequence of batches. We continue with the running example of two batches $\{a, c, f\}$ and $\{a, b, h\}$, the synchronous and asynchronous processing of which has previously been described in Section 2.1.

**Parameter Timestamps**   To achieve equivalence to synchronous execution, we first introduce timestamps associated with each CPU parameter as highlighted with shaded red in Figure 4a in the Appendix. Parameter timestamps are used to determine the correct parameter value in the presence of multiple versions. As shown in Figure 4a, reading batches now requires reading the parameter timestamps as well. Critically, to write a parameter value, a thread must first ensure that it has a larger timestamp for that parameter than the value it is trying to overwrite. Smaller timestamps are not allowed to overwrite larger timestamps. The importance of this point will be highlighted in the example below.

Timestamps are initialized to minus one.

**Parameter Locks**  We also introduce per parameter locks which must be acquired before reading or writing a parameter and its corresponding timestamp (shown graphically with the red outlined rectangle of parameter $a$ in Figure 4a). This ensures each read and write of a parameter value plus its timestamp is atomic, and is needed in the presence of concurrent reader/writer threads (a parameter may be more than just a single float for example, thus reading or writing it may be more than just one operation).

Given parameter timestamps and locks, training proceeds as depicted in Figure 4a. As was the case in asynchronous training, batches can be read and transferred to the GPU in parallel. The key difference between OAC and and asynchronous training comes during GPU batch processing.

**Validation**  When a batch is removed from the GPU input queue for computation, OAC requires that it first check a new on GPU cache of parameter values. This cache serves to track the parameters accessed by concurrent batches and allows each batch to ensure it has the correct values according to synchronous training with the computation order.

In our running example, batch $\{a, c, f\}$ is first to be processed by the GPU. As shown in Figure 4a, it checks the cache but finds no information as it is the first batch in the computation order. The batch then proceeds with model training and updates its values $\{a, c, f\} \rightarrow \{a', c', f'\}$. After updating the values in a batch, the GPU also assigns these updates a new timestamp according to a global GPU timestamp counter. The global timestamp is then incremented. In Figure 4a the updates $\{a', c', f'\}$ are assigned timestamp zero. Assigning timestamps to parameters immediately after they are update by the GPU means timestamps capture the computation order. Finally, before entering the GPU output queue to be transferred back to the CPU, updated parameter values and their timestamps are written to the GPU cache. Batches which were read and transferred in parallel can then subsequently validate against these updates.

The other batch in our running example, $\{a, b, h\}$, will subsequently be processed by the GPU. Thus, it is second in the computation order.  We must ensure that our processing of this batch is equivalent to the computation that would have occurred had we waited for the updates $\{a, c, f\} \rightarrow \{a', c', f'\}$ to make it back to the CPU. Processing of this batch is depicted in Figure 4b. As above, it must validate against concurrent batches by checking the GPU cache. In this case, it detects a conflict for parameter $a$ versus $a'$. A concurrent batch has already modified this value. The timestamp of $a'$ in the cache is zero while the batch has timestamp minus one for its version of $a$. Thus,

it knows the value of $a'$ is the more recent version for this parameter according to the computation order.  It must replace $a$ with $a'$ before it proceeds with computation. As such, the batch $\{a, b, h\}$ is first updated in the validation phase to $\{a', b, h\}$, and then updated in model training to $\{a'', b', h'\}$.  After computation this batch is assigned timestamp one, and its updates are added to the cache. The value for $a'$ with timestamp zero is replaced by the newer timestamp version of this parameter $a''$. The resulting cache state is shown in Figure 4c.

To complete our running example, batches can be transferred back to the CPU in parallel.  They can also write their updates to CPU memory in parallel, but must grab per parameter locks and obey the timestamp rules described above. If the batch with timestamp one acquires the lock to parameter $a$ first, it will update $a \rightarrow a''$ and update the timestamp to one. When the batch with timestamp zero subsequently acquires the lock to parameter $a$ (now $a''$), it will notice that the CPU timestamp is larger than its timestamp. Thus it must not overwrite this value. It is required to do nothing and release the lock. This ensures that older updates according to the computation order do not overwrite newer updates. Notice that the final state of our CPU parameters is $\{a'', b', c', d, e, f', g, h'\}$—this is equivalent to the final state achieved with synchronous training using this order of batches (as described in Section 2.1).

### 3.3. Implementation

The main algorithmic details behind OAC have been presented in the previous section.  Here we discuss several considerations required to implement OAC in practice.

**Cache Eviction**  While we have discussed adding parameter values and their timestamps to the on GPU validation cache, we have not yet discussed when we can evict values. Eviction is required to prevent the cache size from growing indefinitely, as it is assumed in the problem setting that the full set of parameters is too large to fit in GPU memory.

A parameter can be evicted when we can ensure that all subsequent batches to be processed by the GPU will have already had the chance to see this parameter value when they were prepared on the CPU. In other words, a parameter with timestamp $x$ on the GPU can be evicted when we can guarantee that future batches seen by the GPU which contain this parameter will have read a timestamp greater than or equal to $x$ for this value from the CPU memory.

In OAC, we utilize additional metadata to help implement cache eviction. First, the CPU tracks the *maximum finished sequential batch timestamp (MFT)* received from the GPU. This value is updated atomically after each batch finishes writing. It corresponds to the largest continuous batch times-

tamp which has finished. For example, if batches with timestamp $\{0, 1, 2, 5, 6\}$ have completed, regardless of the order in which they finished, the maximum finished sequential batch timestamp is 2. If batches 3 and 4 later arrive, the MFT will then be 6. Just before a batch is read, this value is read atomically and added to the batch metadata. *In this way, each batch knows the maximum timestamp for which it can guarantee that all updates equal or prior to this value in the computation order were present in the CPU memory when it began reading parameter values.* Thus, a batch with MFT equal to $y$ will not need to read parameter values with timestamp less than or equal to $y$ from the GPU validation cache. Note that this does not mean that all individual parameters in the batch have timestamp greater than or equal to $y$, as not all parameters are updated for every batch. It does mean, however, that if a parameter $a$ in this batch has timestamp $y - 1$, for example, then the batch with timestamp $y$ did not update parameter $a$ (otherwise the value read from CPU memory for $a$ would have had timestamp at least $y$). Therefore it is still correct that no parameter values with timestamp less than or equal to $y$ will be read from the GPU validation cache for this batch, as it is impossible for parameter $a$ to find an update with timestamp value $y > y - 1$ on the GPU.

The final metadata required for cache eviction is a dictionary of outstanding batches and their MFT values. Upon creation, each batch is assigned an ID (not necessarily the same as its computation order timestamp). The pair $\{BatchID : BatchMFT\}$ is atomically added to an *outstanding batches* dictionary. When batches finish writing, their key-value pair is atomically removed from this data structure. For cache eviction, when a batch reads its MFT value and adds its key-value pair to the outstanding batches dictionary, it also atomically calculates the minimum MFT value across all entries in the dictionary. This value is stored in the batche's *safe to evict (STE)* metadata field. When batches are read by the GPU, all parameter values with timestamp less than or equal to the STE metadata can be evicted. The GPU performs this eviction for each batch.

When a batch calculates its safe to evict timestamp, even if this batch somehow beats all other outstanding batches to the GPU, it is still okay to perform eviction as described above. This is because all subsequent outstanding batches have an MFT greater or equal to the STE value. That means that they were prepared on the CPU after the entries we wish to evict had already been persisted in CPU memory. Any future batches that will be prepared on the CPU will also have an MFT greater than or equal to this value. Thus no batch will ever again reach the GPU and find a timestamp for a parameter less than the STE value but greater than the timestamp it read from CPU memory. Therefore, no batch will ever again need to read these values from the cache and they can be safely evicted.

**Deadlock Prevention** When implementing OAC, one also needs to consider whether the per parameter locks introduced for ensuring atomic reads and writes to parameter values and their timestamps can introduce deadlocks. In principle the answer is yes, however we have not observed this phenomena in practice. One possible reason is that these locks are extremely lightweight. They are held only for the time it takes to read or write a few bytes from CPU memory. For this reason, currently we do nothing to prevent deadlocks in our OAC implementation. One can simply detect deadlocks by monitoring the GPU utilization (which would drop to zero if no batches are making progress), and if a deadlock is detected, simply revert the parameters to the last checkpoint and begin training again.

If deadlocks were prevalent, a simple solution would be to require that all batches read and write the parameters in a global order. For example, this could be achieved by assigning the parameter values IDs and then reading and writing in sorted ID order.

## 4. Evaluation

In this section, we evaluated our proposed method OAC by comparing its throughput and convergence with standard synchronous and asynchronous training. We find that OAC achieves its desired goal: synchronous convergence while maintaining asynchronous throughput, leading to the best time-to-accuracy of the three methods.

### 4.1. Experimental Setup

As described in Section 2.2, we implement our OAC prototype in the graph learning system Marius, and test on the graph learning task of link prediction. We use the Freebase86m knowledge graph (Google, 2018). This graph has roughly 86 million nodes, and we learn 50-200 parameters (called the embedding dimension) for each node, depending on the model. This leads to 17-68GB of model parameters (each parameter is four bytes). We store these in CPU memory. We measure our link prediction model quality using the metric Mean Reciprocal Rank (MRR) (Mohoney et al., 2021). This quantity tries to capture how well learned parameters can recover the edges of the graph and takes values between zero and one—the higher the better. We run experiments on two machines. One with 80 CPU cores and one with 20 CPU cores. Both machines have one NVIDIA Tesla V100 GPU with 32GB RAM. Note that even when the model is only 17GB, we can not store all of it in the GPU, as we must also store the edges (3.6GB), and leave enough GPU memory for the model to perform the computation required for training.

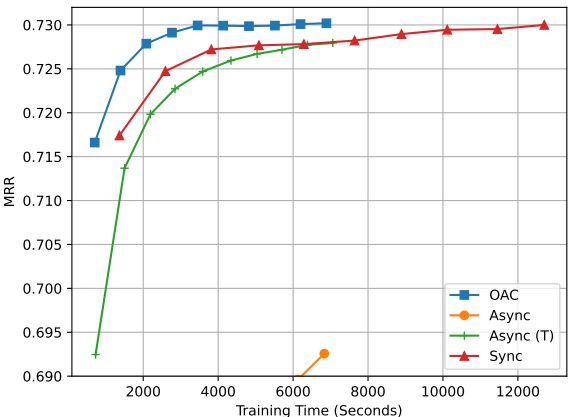

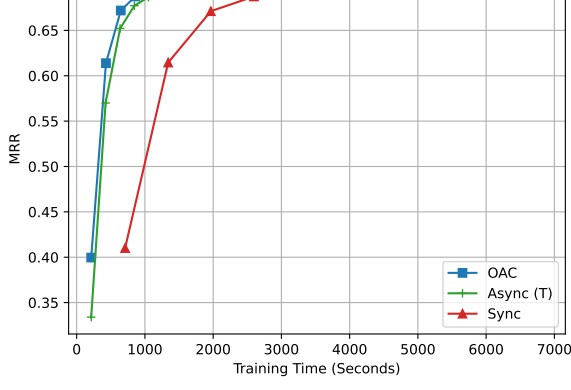

(a) Setup 1: We train a GraphSage (Hamilton et al., 2017) plus DistMult (Yang et al., 2014) model with embedding dimension 50 using ten neighbors and a batch size of 50k. Training was performed on a machine with 80 CPU cores and one NVIDIA Tesla V100. Each batch accesses roughly 100 million parameters.

(b) Setup 2: We train a DistMult model with embedding dimension 50 using a batch size of 500k. Training was performed on a machine with 20 CPU cores and one NVIDIA Tesla V100. Each batch access roughly 50 million parameters.

*Figure 2.* Time-to-accuracy for OAC versus asynchronous and synchronous training.

## 4.2. Time-To-Accuracy

As described in the introduction, the primary practical metric of interest for comparing OAC with synchronous and asynchronous training is time-to-accuracy. In Figure 2 we plot the MRR versus wall clock training time for the three methods using two different setups. Each method is trained for ten *epochs* and we measure the MRR after each one. Here, an epoch refers to one full pass over the training examples and results in a fixed number of batches. Thus all methods see the same total number of batches pass through the GPU.

In Figure 2a, asynchronous training is shown by the orange line (labeled Async) barely visible in the bottom middle of the plot. For this setting, we maximize the number of reader and writer threads and queue size parameters to maximize throughput (these parameters were discussed in Section 2.1). In this case, blindly running asynchronous training causes a severe degradation in model quality. We also plot a tuned version of asynchronous training (Async (T)), where we have manual fiddled with the number of threads and queue sizes to find the best configuration. This allows us to regain much of the lost accuracy of asynchronous training with very little throughput loss. Both Async and Async (T) finish the allotted number of batches roughly twice as fast as synchronous training (roughly 6500s versus 12500s), but synchronous training ends with a higher MRR. OAC outperforms all other methods and does not require the manual tuning of Async (T). We simple maximize the reader/writer threads and queue sizes but use our method described in Section 3. OAC achieves the fastest time-to-accuracy. This

is because it maintains the throughput of asynchronous training, also finishing the 10 epochs in 6500s, while guaranteeing the convergence of synchronous training (which we highlight in the following section).

Figure 2b shows a second training setup. In this case, tuned asynchronous training manages to achieve similar accuracy to synchronous training while learning more than three times faster. We do not show the default asynchronous training as it is roughly equivalent to the tuned setup. The reason asynchronous training is able to performs well in this case is twofold: First, batches access roughly half the number of parameters of the setup in Figure 2a. Second, the machine has one quarter of the CPU resources, limiting the number of batches it can process concurrently. Both of these differences result in fewer concurrent batches which overlap—recall that the fewer conflicts there are, the more likely asynchronous training is to perform well. That said, however, in Figure 2b OAC also trains over three times faster than synchronous training and is always guaranteed to reach the same model quality.

## 4.3. Convergence

To highlight that OAC converges at the same rate as synchronous training, we plot the same experiments from Section 4.2/Figure 2 again in Figure 3, but this time versus epoch number rather than versus training time. Recall that for each epoch, all methods see the same number of batches pass through the GPU, thus Figure 3 shows the accuracy with respect to the number of processed batches. We have already shown in Section 4.2 that asynchronous training

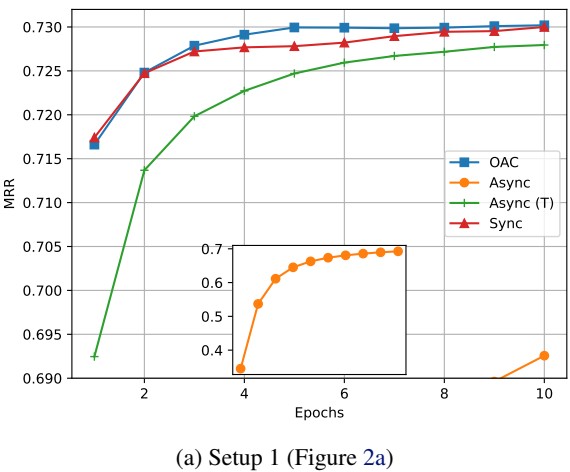

(a) Setup 1 (Figure 2a)

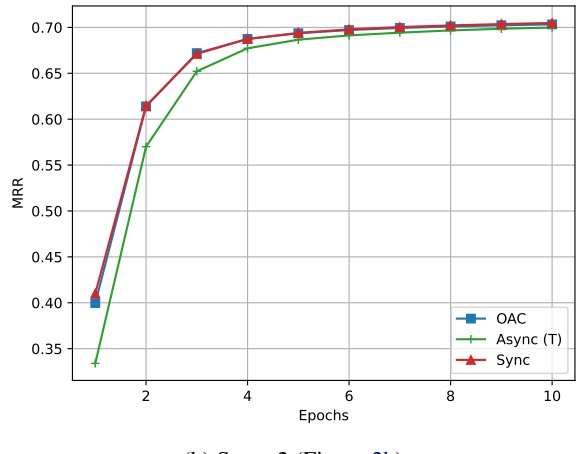

(b) Setup 2 (Figure 2b)

*Figure 3.* Convergence of OAC versus asynchronous and synchronous training for the experimental setups described in Figure 4.2. See Section 4.3 for a discussion.

and OAC have higher throughput than synchronous training (processing more batches per unit time).

Figure 3 shows that OAC matches the convergence of synchronous training. In Figure 3a, asynchronous training converges so poorly, that it requires its own scaling of the vertical axis. Tuned asynchronous training converges faster, reaching 0.6925, 0.725, and 0.7275 MRR after 1, 5, and 10 epochs respectively. Synchronous training and OAC both achieve 0.7175 MRR after one epoch and 0.73 MRR after 10 epochs. In this case, OAC converges slightly faster than synchronous training during the middle epochs. This is because OAC is guaranteed to be equivalent to *some* synchronous order, not necessarily the exact synchronous order given by the red line in Figure 3a.

The convergence of each method for the training setup of Figure 2b is shown in Figure 3b. As described above, in this case asynchronous training actually performs quite well—it converges at nearly the same rate as synchronous training. Again, however, OAC is guaranteed to converge at the rate of some synchronous order. In Figure 3b, the OAC and synchronous lines nearly perfectly overlap.

## 5. Conclusions

In this work, we have introduced Optimistic Asynchrony Control (OAC), a method for allowing concurrent processing and transfer of batches in a mixed CPU-GPU ML training setup while ensuring that the final result is equivalent to a serial, synchronous execution. The key idea was to allow all batches to run in parallel, but to validate each batch against concurrent batches immediately before computation on the GPU. This allowed us to ensure that the correct (according to a synchronous order) parameter values were

used for model learning. We have shown that OAC maintains the throughput of asynchronous ML training while converging at the rate of synchronous machine learning. As a result, OAC can achieve the best time-to-accuracy and requires no manual tuning. OAC is likely to be most useful for applications where asynchronous training is rendered unusable because it results in unacceptably lower final model accuracy. Due to the success of this work, we will likely implement OAC in the public version of Marius, making it available to all users who wish to train graph learning models.

## Acknowledgements

We would like to thank the anonymous reviewers for their constructive comments that will help us improve our work. We would also like to thank Professor Xiangyao Yu who provided feedback and suggestions on this work when it was a class research project in CS 764 at UW-Madison.

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

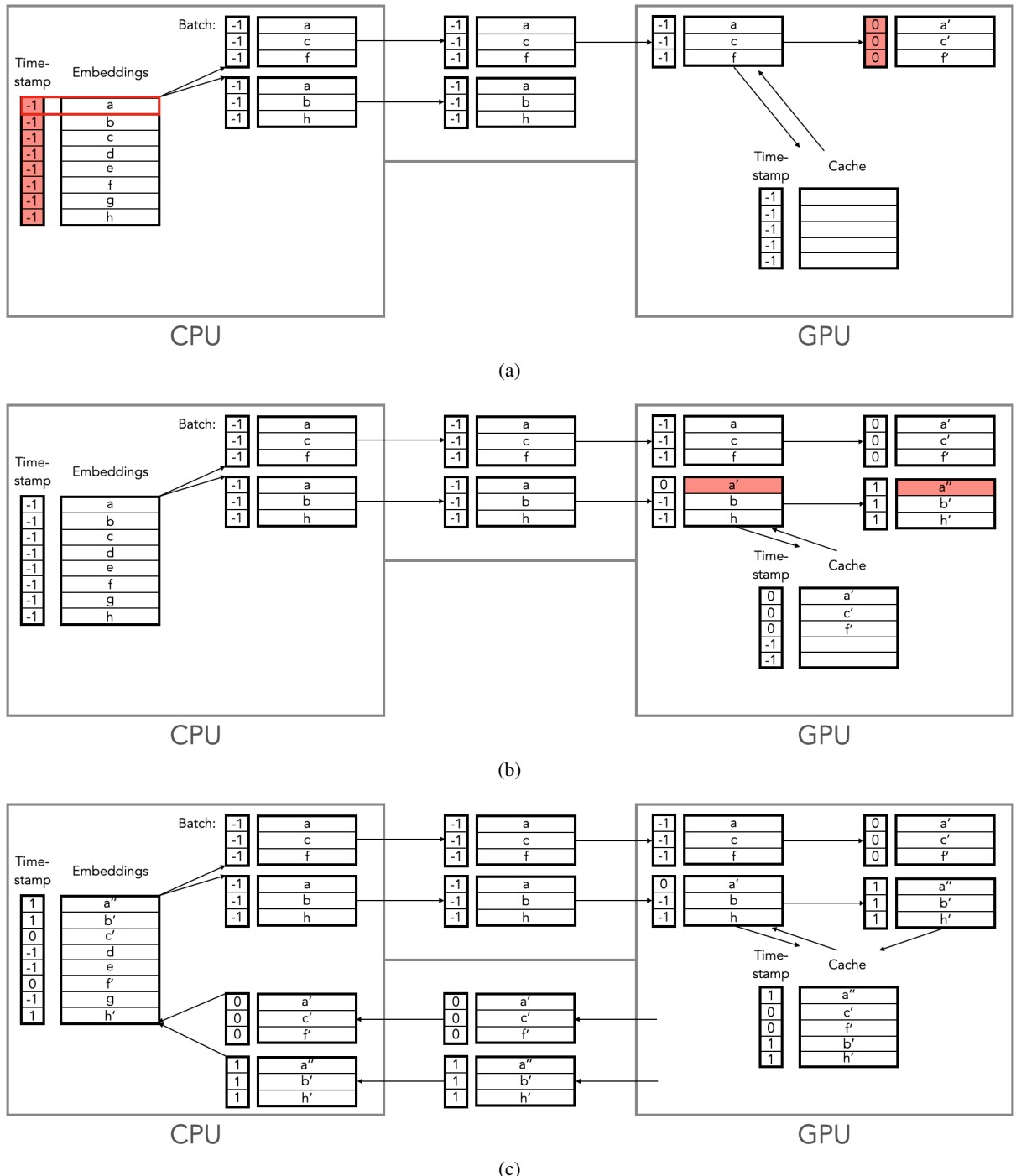

*Figure 4.* Depiction of OAC parallel processing of batches while ensuring equivalence to a synchronous execution. Details are discussed in the text of Section 3.2.

