# OpenReview forum: "Optimistic Asynchrony Control: Achieving Synchronous Convergence With Asynchronous Throughput for Embedding Model Training"
_ICML.cc/2024/Workshop/WANT — WANT@ICML 2024 Poster_

### Official Review · Reviewer_oS4U · 2024-06-07
**Asynchronous Training with Parameter-Update Order Guarantees**

**Confidence:** 4

**Summary:**

This paper introduces Optimistic Asynchrony Control (OAC), an algorithm for batching training data, using parallel batch transfers and optimistic locking. OAC takes inspiration from optimistic concurrency control methods used in concurrent databases and applies this concept to model training, with the aim of improving model training times, particularly of large models on CPU-GPU systems with significant memory limitations. In the paper, authors describe the workings of the algorithm and provide experimental results.

**Strengths:**

* Compelling experimental results show that OAC performs better than some traditional asynchronous training methods, and has properties on par with synchronous training, which supports their theoretical assertions.
* The problem being solved, of efficiently coordinating CPU-GPU training with memory limitations, is quite relevant.
* The concept of applying optimistic locking to model parameters is sound, and seems to have been implemented carefully.
* Good discussion of technical details associated with implementing the algorithm.

**Weaknesses:**

* Explanation of the OAC algorithm and its correctness is quite long-winded and relies primarily on small, worked examples.
* No source code available, however it was mentioned that OAC will be added to the open source program Marius, which is promising.
* Only two experiments performed.

**Suggestions:**

* A more concise justification, using more general arguments, of why OAC performs the same parameter updates as synchronous training.
* Pseudocode for core parts of the algorithm, such as management of timestamps and parameter locks.
* More reference and discussion of existing optimistic concurrency control methods, and how OAC relates or differs.
* It would be good to see more experiments done with a variety of learning tasks and models.
* Line 371: "sever" should be "severe".

---

### Official Review · Reviewer_o4md · 2024-06-12
**Review for paper "Optimistic Asynchrony Control"**

**Confidence:** 3

**Summary:**

This paper proposes a mechanism for asynchronous batch training system between main and accelerator memory, based on Optimistic Asynchrony Control.
Specifically, the authors validate each batch before updates are committed to ensure that updates are atomically altered. Moreover, they provide a caching mechanism on the GPU to enhance throughput.
Evaluation on two graph embedding problems showcase superior time-to-accuracy results compared to the synchronous and asynchronous baselines.

**Strengths:**

* The system is clearly motivated and the parallelism to DBMSs seems to intuitively make sense.
* The authors implementation on top of a largely used system (Marius) is a positive aspect. However, open-sourcing their prototype would enhance reproducibility.
* The results of the evaluation showcase a measurable benefit of this techique.

**Weaknesses:**

* The paper is overly long in its background, with too much repetition until page 3.
* The overhead of book-keeping has not been measured, which can be sizeable
* The baselines to which the system is compared only have the two ends of the spectrum.

**Limitations:**

* Results of the evaluation do not include variance statistics, which may suggest that experiments were only run once.
* I am not sure whether this technique can generalise across multiple accelerators (and memory hierarchies). At the same time, I am unsure how this technique can be integrated with systems that support RDMA.
* The current system is mainly applied on graph problems, but the title or abstract do not make this specific.
* The authors propose a mechanism for validating the updates to parameters without quantifying the cost of running this. From the information provided, the overhead should be sizeable.
* The authors keep a timestamp per model parameter. This can be detrimental in terms of overall memory consumption. If we add optimiser state parameters, this can further deteriorate.
* The paper states that each batch is validated against every other concurrent batch, which would indicate a quadratic cost.
* It is unclear from the paper how much the conflict rates affect the performance of the system.
* It is further unclear what is the memory overhead of the cache.

**Suggestions:**

* I would suggest that certain terms are changed:
    - embedding models -> representation learning
    - CPU memory -> main memory
* I would urge the authors to also discuss to the energy impact of sync vs. async training.
* The paper would benefit from an algorithmic representation of the system's workflow.

---

### Official Review · Reviewer_UTxx · 2024-06-14
**Thoughtful algorithmic improvement; tricky implementation; limited experimentation.**

**Confidence:** 4

**Summary:**

The paper introduces an asynchronous optimization algorithm (communication and batch processing are async) for representation learning of word models, graph models, or any other knowledge models. The main contribution is thoroughly design batch cache on GPU-side which provides operations ordering as if all batches were processed synchronously. In order to guarantee proper processing order, the authors introduce vector clock which assigns a timestamp to each training sample (representation vector). Proposed method was verified empirically and demonstrated advatage over synchronous algorithms and asynchronous ones without ordering enforcing.

**Strengths:**

Thoroughly design, described in details, and instructive asynchronous algorithm which is determenistic with respect to synchronous counterpart and has better performance and efficiency.

**Weaknesses:**

+ Bibliography should be reviewed and actualized: capitalization of titles,
  missing publication dates, journals conferences, etc (e.g.
  (Kipf&Welling,2022) has been published in ICLR
  https://openreview.net/forum?id=SJU4ayYgl)
+ Missing table of content in hypertext markup.

+ Some pieces are hard to follow. Please rephrase sentences like below or use
  punctuation to logically split parts of the utterance.

  > We must ensure that our processing of this batch is equivalent to the
  > computation that would have occurred had we waited for the updates {a, c, f
  > } → {a0 , c0 , f 0 } to make it back to the CPU.

+ Implementation details are completely unclear since cache eviction procedure
  is quite complex. Specifically, it is unclear how exactly the authors
  implemented GPU cache. Do they used like two buffers for the new and old
  batches? Do they run a specialized CUDA-kernel to merge caches?

  Source code could solve the issues if it were attached to the submission.

+ I belive that there is a better experimental setup than link prediction task.
  From my perspective, it would be more representative to compare with word2vec
  since it is the first algorithm which introduced the notion of async
  optimization despite that the original implementation is CPU-only. Also,
  word2vec has more sophisticated evaluation protocol which is evaluation of
  similarity of word representations in addition to synthetic metrics.

+ The latter point raises another question related to locks and ordering.
  Despite the fact word2vec algorithm does not uses any locks to access model
  weights, it convergence to meaningful representations. Maybe, we do not need
  properly ordered async updates in practice? In this perspective, Figure 2
  does not look convincing to me thus more experimentations in different
  setups are required.

**Suggestions:**

By design, the algorithm is limited to a specific models for training vector representation. However, it seems that the algorithm could be adopted for rather complex models like Transformers in PEFT setup or MoE models as an example. How do you think?

---

### Decision · Program_Chairs · 2024-06-18

**Decision:**

Accept (Poster)

**Comment:**

We thank the authors for their time and contribution to WANT and we are pleased to share that after the reviewing process the paper has been accepted. Congratulations! We encourage the authors to consider reviewers' feedback for the improvement of the camera-ready version. We hope to see you in person at the workshop and brainstorm on efficient training research together!